# Effectiveness of a Multifaced Antibiotic Stewardship Program: A Pre-Post Study in Seven Italian ICUs

**DOI:** 10.3390/jcm11154409

**Published:** 2022-07-28

**Authors:** Giulia Mandelli, Francesca Dore, Martin Langer, Elena Garbero, Laura Alagna, Andrea Bianchin, Rita Ciceri, Antonello Di Paolo, Tommaso Giani, Aimone Giugni, Andrea Gori, Ugo Lefons, Antonio Muscatello, Carlo Olivieri, Angelo Pan, Matteo Pedeferri, Marianna Rossi, Gian Maria Rossolini, Emanuele Russo, Daniela Silengo, Bruno Viaggi, Guido Bertolini, Stefano Finazzi

**Affiliations:** 1Istituto di Ricerche Farmacologiche Mario Negri IRCCS, 20156 Milano, Italy; giuliamandelli.bg@gmail.com (G.M.); elena.garbero@marionegri.it (E.G.); guido.bertolini@marionegri.it (G.B.); stefano.finazzi@marionegri.it (S.F.); 2Associazione GiViTI—Gruppo Italiano per la Valutazione degli Interventi in Terapia Intensiva, 24020 Ranica, Italy; 10mlanger@gmail.com (M.L.); r.ciceri@asst-lecco.it (R.C.); aimonegiugni@gmail.com (A.G.); carlo.olivieri@aslvc.piemonte.it (C.O.); m.pedeferri@asst-lecco.it (M.P.); daniela.silengo@gmail.com (D.S.); bruno.viaggi@gmail.com (B.V.); 3Emergency-Ong, 20128 Milano, Italy; 4Infectious Diseases Unit, Fondazione Istituto di Ricovero e Cura a Carattere Scientifico (IRCCS) Ca’ Granda Ospedale Maggiore Policlinico, 20122 Milan, Italy; laura.alagna@gmail.com (L.A.); andrea.gori@unimi.it (A.G.); antonio.muscatello@policlinico.mi.it (A.M.); 5Anesthesia and Intensive Care, Ospedale Civile San Valentino di Montebelluna, 31044 Montebelluna, Italy; andbia@libero.it; 6Anesthesia and Intensive Care, Ospedale Alessandro Manzoni di Lecco, 23900 Lecco, Italy; 7Department of Clinical and Experimental Medicine, Università di Pisa, 56126 Pisa, Italy; antonello.dipaolo@unipi.it; 8Department of Experimental and Clinical Medicine, Università di Firenze, 50134 Firenze, Italy; tommaso.giani@unifi.it (T.G.); gianmaria.rossolini@unifi.it (G.M.R.); 9Clinical Microbiology and Virology Unit, Azienda Ospedaliero Universitaria Careggi, 50134 Firenze, Italy; 10Department of Intensive Care and Emergency Medical Services, Ospedale Maggiore, 40133 Bologna, Italy; 11Department of Pathophysiology and Transplantation, Università degli Studi di Milano, 20122 Milan, Italy; 12Centre for Multidisciplinary Research in Health Science (MACH), Università degli Studi di Milano, 20122 Milan, Italy; 13Anesthesia and Intensive Care, Ospedale Alta Val d’Elsa di Poggibonsi, 53036 Poggibonsi, Italy; ugo.lefons@uslsudest.toscana.it; 14Anesthesia and Intensive Care, Ospedale Sant’Andrea, ASL VC Vercelli, 13100 Vercelli, Italy; 15Infectious Diseases Unit, Istituti Ospitalieri di Cremona, 26100 Cremona, Italy; angelo.pan@asst-cremona.it; 16Anesthesia and Intensive Care, Presidio Ospedaliero San Leopoldo Mandić, 23807 Merate, Italy; 17Division of Infectious Diseases, “San Gerardo” Hospital, University of Milano-Bicocca, 20900 Monza, Italy; marianna.rossi@asst-monza.it; 18Anesthesia and Intensive Care, Ospedale Maurizio Bufalini di Cesena, 47521 Cesena, Italy; lelegaiola@gmail.com; 19Anesthesia and Intensive Care, Ospedale San Giovanni Bosco, 10154 Turin, Italy; 20Neuro Intensive Care Unit, Department of Anesthesiology, Azienda Ospedaliero Universitaria Careggi, 50134 Firenze, Italy

**Keywords:** antibiotic stewardship, multidrug resistance, intensive care units, healthcare-associated infections, infection control, electronic health record, education in medicine, appropriateness of antibiotic

## Abstract

Multidrug resistance has become a serious threat for health, particularly in hospital-acquired infections. To improve patients’ safety and outcomes while maintaining the efficacy of antimicrobials, complex interventions are needed involving infection control and appropriate pharmacological treatments in antibiotic stewardship programs. We conducted a multicenter pre-post study to assess the impact of a stewardship program in seven Italian intensive care units (ICUs). Each ICU was visited by a multidisciplinary team involving clinicians, microbiologists, pharmacologists, infectious disease specialists, and data scientists. Interventions were targeted according to the characteristics of each unit. The effect of the program was measured with a panel of indicators computed with data from the MargheritaTre electronic health record. The median duration of empirical therapy decreased from 5.6 to 4.6 days and the use of quinolones dropped from 15.3% to 6%, both *p* < 0.001. The proportion of multi-drug-resistant bacteria (MDR) in ICU-acquired infections fell from 57.7% to 48.8%. ICU mortality and length of stay remained unchanged, indicating that reducing antibiotic administration did not harm patients’ safety. This study shows that our stewardship program successfully improved the management of infections. This suggests that policy makers should tackle multidrug resistance with a multidisciplinary approach based on continuous monitoring and personalised interventions.

## 1. Introduction

The efficacy of antimicrobials still saves the vast majority of patients suffering from bacterial or fungal infections. However, their use, overuse and mainly inappropriate use in and outside hospitals, as well as in livestock, favours the emergence of resistance. Resistant bacterial species threaten health and cause related morbidity and even mortality [1]. This has become a general emergency in hospitals and in general medical practice—although with significant geographical differences [2]. However, it is recognised that judicious use of antimicrobials is a cornerstone of the containment of multidrug resistance (MDR) [3]. 

Antibiotic stewardship programs (ASPs) are accepted worldwide as a must to improve patients’ safety and outcomes, while maintaining the efficacy of antimicrobials by withholding the selective pressure driving antibiotic resistance (ABR) [4]. ASP comprises a bundle of interventions to improve several aspects of a complex decision-making process [5] involving organisation, prevention of transmission, diagnosis of infection, handling of microbiological investigations, optimisation of drug prescriptions [6], and duration of treatments.

There is general agreement on the urgent need for effective ASP, the best bundle composition, and the best way to implement these programs and to maintain the benefit over time. Most published stewardship programs, using very different methods, report success in achieving specific goals [7,8,9,10,11,12,13,14]. However, better management of infections calls for the design and achievement of several goals: reduction of the circulation and transmission of MDR [15] microorganisms and more appropriate use of drugs (sparing of carbapenems, limitation of quinolones and other broad-spectrum drugs, and appropriate site, dose, and duration of treatments).

Intensive care units (ICUs) present unique challenges for ASP due to their crucial position in the chain of antibiotic resistance: they admit critical and chronically ill patients frequently colonised by MDR microorganisms, transferred from hospital wards and nursing homes [16]. ICU doctors use antimicrobials generously, and return survivors with a greater or even unit-acquired MDR burden to the hospital and the community [17]. However, ICU personnel, having experienced how difficult it is to treat patients with MDR infections, do frequently pay closer attention to the MDR problem. ASPs have often been optimised in ICUs in recent years, with attempts also to develop the multidisciplinary aspect by including infectious diseases, microbiologists, and pharmacists in the projects.

In 2017 the Italian Group for the valuation of Intervention in Intensive Care Units (GIViTI, giviti@marionegri.it) started a multi-ICU project to control antibiotic resistance through a complex peer-to-peer intervention and extended monitoring with a common electronic health record (EHR), MargheritaTre (M3) [18] as a potential continuous antibiotic-stewardship tool.

The aim of this before/after project, intended as a pilot study, was to assess the efficacy of an ASP in a multicenter study. Specific goals of the ASP were reduction of the overall antibiotic pressure, sparing of the essential anti-MDR-drugs (e.g., carbapenems, colistin, linezolid), reduction of the use of quinolones, optimisation of drug administration, and improvement of appropriateness of antibiotic treatment. Appropriateness was assessed across several dimensions, focusing on infections with valid diagnostic specimens, microbiological diagnoses and pharmacologic properties as tissue penetration of the prescribed drugs. These actions, together with prevention of transmission, should yield the very ambitious achievement of reducing MDR infections. Considering the complexity of such a project, the ASP intervention was designed by a multidisciplinary team and agreed with the representatives of the participating ICUs.

The performance of each center was evaluated through a set of indicators designed to monitor several dimensions in the management of infections. The ASP interventions were tailored to each ICU on the basis of data collected during the first year of the project (before the intervention) and discussed with a panel of experts at on-site visits. The impact of the ASP over the year of observation was assessed by comparing the values of the indicators before and after the intervention. A further year of observation was planned to verify how long the benefits, if any, lasted.

## 2. Methods

### 2.1. Study Design

We ran a multicenter pre-post study to assess the impact of a stewardship program in ICUs. The program was based on a plenary meeting with representatives from ICUs and on-site audits. The impact of the program was measured by comparison of a panel of indicators computed before and after the intervention.

### 2.2. Participating Units

Participation in the study was voluntary, but limited to units working with the software M3, integrated with the laboratory information system. M3 is an EHR developed by a multidisciplinary team involving IT specialists, researchers, physicians, and nurses from the GiViTI network. It was designed to support clinical practice in ICUs and ensure high-quality data for research purposes [18]. M3 is property of Istituto di Ricerche Farmacologiche Mario Negri IRCCS (Milano, Italy) and GiViTI (Ranica, Italy). 

### 2.3. Study Population

The study population comprised all patients admitted to seven general Italian ICUs of different sizes and case-mix. The study took place between January 2017 and February 2020 and had 12 months of data collection (see the Appendix A for the list of ICUs and their characteristics). 

### 2.4. Data Collection and Management

All data (clinical and microbiological diagnoses, laboratory tests, and treatments) were automatically acquired from the M3 EHR, without further intervention of the ICU physicians, limiting the risk of biases. 

Information in M3 is primarily stored in structured or partially structured form to facilitate data analysis. Automatic services import patients’ parameters and results of chemical and microbiological tests from monitors, ventilators, blood–gas analyser devices, and from the hospital information systems. M3 stores patients’ data in a local PostgreSQL database in each hospital. Data are then encrypted and transferred in pseudonymised form to a server at the GiViTI coordinating center at the Mario Negri Institute for Pharmacological Research.

For this project we extracted the following variables from M3 databases: present-at-admission or ICU-acquired infection, site of infection, microbiological diagnosis and sensitivity pattern, where available, antimicrobials employed (drug, start and end dates of treatment, drug combinations), the rationale for antibiotic prescription (prophylaxis, targeted or empirical therapy), and length of ICU stay.

### 2.5. Phases of the Project

The study was coordinated and monitored by a study board nominated by the GiViTI steering committee. The members of the board were chosen for their expertise in critical care medicine, infectious diseases, clinical microbiology, and data science. The board defined the project’s specific objectives and designed all the phases of the intervention.

Definition of the indicators: The study board designed all indicators to measure several dimensions related to the management of infections in ICUs (resistance patterns of the isolated microorganisms to drug classes, appropriateness of drug use, clinical decisions). When needed, the EHR M3 was modified to collect the variables employed to calculate those indicators.

Plenary session: A kick-off meeting was organised with representatives of the ICUs (nurses, intensivists, microbiologists, infectious disease specialists, pharmacists/pharmacologists) to share the objectives of the project, to describe its phases, and to recall and discuss standard strategies for infection control in the ICU and what is known to limit the emergence of antimicrobial resistance. This course was structured with plenary lectures and workgroups based on case records extracted from the EHR of the ICUs. The indicators to describe and quantify the measured data were discussed in this meeting.

To build a common multidisciplinary background, the topics discussed in the meeting aimed to update knowledge about risk stratification, diagnosis of infection, antibiotic prescription for community- and hospital-acquired infections, PK/PD optimisation, interpretation of antibiotic sensitivity tests for classical and novel diagnostic technologies, communication strategies with the laboratories, and information from biomarkers. The importance of environmental cleanliness and prevention of transmission were stressed as fundamental issues

On-site visits and follow up: All ICUs were visited between October 2018 and February 2019 by experienced members of the study board. The multidisciplinary visiting team involved an intensivist, a clinical microbiologist, an infectious disease specialist, and a data scientist from the coordinating team. Each visit lasted a whole day. The morning was dedicated to visiting the ICU and the microbiology laboratory to study the organisation of clinical activities and the decision-making. In the afternoon, pre-intervention data were evaluated, and critical aspects were identified and discussed. In a final de-briefing, the ICU members and the peers agreed on and fixed the goals to be achieved in one year. During this year each ICU could consult the clinical experts.

Final evaluation of the results: One year after the visit data from each center were processed and the indicators computed, each center received a report comparing its own performance to all the other ICUs.

The results were presented in a GiViTI meeting organised in online format due to COVID-19 restrictions in Italy.

### 2.6. Outcomes

The success of the stewardship program was evaluated through the following indicators. Mortality and ICU length of stay were used as safety parameters to make sure that the intervention did not harm patients.

#### 2.6.1. Frequency of Patients with MDR Infections

Ratio of patients with at least one infection due to MDR bacteria according to the definition of Ref. [15] to the total number of infected patients. This endpoint was stratified by infections present at ICU admission or acquired during the ICU stay. Infections whose symptoms appeared during the first 48 h in the ICU were considered infections at admission.

#### 2.6.2. Median Duration of Empirical Therapy and Prophylaxis

Kaplan–Meier curves were built to assess the duration of empirical therapies and prophylaxis (antimicrobial treatments aiming to avoid infections, including perioperative prophylaxis), censoring patients with ongoing therapies at discharge (see Appendix A).

#### 2.6.3. Inappropriateness of Antibiotics by Penetration into the Site of Infection

Ratio of inappropriate antibiotic therapies regarding tissue penetration to the number of antibiotics prescribed, based on a recent systematic review [19]. An antibiotic is considered inappropriate when it cannot reach the site of the infection.

#### 2.6.4. Inappropriateness of Antibiotics by Microorganism Resistance Pattern

Ratio of inappropriate antibiotic therapies to the number of antibiotics prescribed. An antibiotic is considered as inappropriate if the bacteria causing the infection are intrinsically resistant [20] or resistant according to susceptibility tests [21,22,23,24].

#### 2.6.5. Use of Fluoroquinolone Antibiotics

Proportion of patients who received at least one fluoroquinolone.

#### 2.6.6. Inappropriate Prescriptions of Carbapenems

Ratio of inappropriate prescriptions of carbapenems to the total number of treatments with these drugs. Treatment with carbapenems is considered inappropriate when the microorganism causing the infection was responsive to other molecules with a more limited spectrum or anti-MDR specificity such as penicillin or cephalosporins.

#### 2.6.7. Inappropriate Prescriptions of Colistin

Ratio of inappropriate prescriptions of colistin to the total number of treatments with these drugs. Treatment with colistin is considered inappropriate when the microorganism causing the infection was responsive to penicillin, cephalosporins, and carbapenems.

#### 2.6.8. Inappropriate Prescriptions of Linezolid

Ratio of inappropriate prescriptions of linezolid to the total number of linezolid therapies. Empirical therapies in patients with acute renal failure were considered appropriate. Therapies in patients with SNC infection by Gram + bacteria or any infection due to MRSA or VRE were deemed appropriate.

The board of experts used three additional indicators to condense the results (before and after) in each ICU concerning patients’ outcomes and drugs used. 

-Antibiotic pressure: Proportion of days of ICU stay when patients received any antibiotic therapy.-Average ICU length of stay.-ICU mortality.

### 2.7. Statistical Analysis

Categorical variables are reported as frequency and percentage, continuous variables as mean and standard deviation (SD) or median and interquartile range (IQR), as appropriate.

Chi-squared and Wilcoxon rank-sum tests were applied to compare proportions and distributions of continuous variables, respectively, with a significance level of 0.05.

To take into account stratification by ICU, the results of the indicators before and after the intervention were compared using Cochran–Mantel–Haenszel, stratified Mann–Whitney, and stratified log-rank tests for proportions, distribution of continuous variables, and Kaplan–Meier curves, as appropriate, with a significance level of 0.05.

All analyses were done with R, version 3.6 (R Core Team, R Foundation for Statistical Computing, Vienna, Austria).

## 3. Results

The indicators were evaluated for data collected in 2018 on 2901 patients to assess the performance of the seven ICUs before the ASP intervention. The program’s efficacy was assessed by comparing the same indicators on data collected for a whole year after the site visits for 3389 patients. The patients’ main characteristics are reported in Table 1.

The indicators computed in the pre- and post-intervention phases are compared in Table 2. Improvement was obtained on the frequency of infections caused by MDR bacteria (39.5% post-intervention vs 44.9% pre-intervention), especially for ICU-acquired infections (48.8% vs. 57.7%). The frequency of MDR in infections on admission and acquired in ICU for each center are plotted in Figure 1a,b, before (dashed) and after (solid) the ASP intervention. The horizontal lines indicate the overall average. Although the Cochran–Mantel–Haenszel tests are not significant, the changes are substantial and the percentage of MDR in ICU-acquired infection decreased in all but one of the participating ICUs.

The median duration of empirical therapy and prophylaxis was reduced from 5.6 to 4.6 days (*p* < 0.001) and from 2.3 to 2.0 days (*p* = 0.06), respectively. The median duration of empirical therapy before the intervention ranged from about 4 to 8 days in the seven ICUs. This decreased in all the ICUs, significantly in four of them (Figure 2a). Regarding prophylaxis, the behaviour of the ICUs differed widely (Figure 2b). The two ICUs with the longest durations before the intervention improved their performance, coming close to the average of all the centers. The duration of prophylaxis significantly increased only in one ICU, nonetheless remaining well below the average.

The only indicator that significantly increased was the use of linezolid, though with a limited number of prescriptions. After our ASP, 69.8% of linezolid prescriptions were inappropriate (as defined in Section 2), while 54.9% were considered inappropriate before the intervention. This worsened in more than half of the centers (Figure 3b), but the confidence intervals are quite wide since only a few patients received linezolid.

How far appropriateness is concerned, 16,2% and 17,3% of empirical, and 3,8% and 4.8% of targeted treatments were considered inappropriate according to our definitions and no pre/post change could be found (Table 2). 

The average length of stay increased (not significantly) from 5.4 to 5.5 days (*p* = 0.07) and mortality remained unchanged (from 16.2% to 15.9%).

The other indicators did not change significantly. They are plotted in the Appendix A.

## 4. Discussion

Continuous education and monitoring and improvement of the quality of care in ICUs are the primary missions of the GiViTI group. Given the lasting interest in the epidemiology and reduction of infectious complications in critically ill patients [25], an ASP study was mandatory. The objectives of an ASP are the containment of infections, better use of antimicrobials, and reduction of the emergence and spread of MDR bacteria. Although these goals are universally recognised, standardised methods for their implementation and monitoring are far from being defined yet.

Here, we report a pilot ASP that was education- and culture-based, with no additional workload or formal protocols for healthcare workers. Its implementation was adapted to the different operating conditions of each ICU. The indicators used to monitor the ICU performance are simple, easy to understand and offer a possible tool for continuous surveillance.

Monitoring was made easier by taking data directly from the EHR M3, thus minimising the risk of bias due to the manual input into an ad hoc case report form. Standardised indicators addressing several items in the ASP were automatically computed from M3 data: admission of infected and MDR-infected patients, ICU-related acquisition of MDR infection, duration of antimicrobial treatments (targeted, empirical, or prophylactic), and number of treatments with specific antimicrobials (carbapenems, colistin, quinolones, and linezolid). 

Outcomes such as the length of stay and mortality cannot be seen as indicators of efficacy but as an attempt to monitor safety. The possibility of benchmarking results in time with a before/after analysis and among units stimulates them to improve their performance and shows that improvements are possible in clinical practice. 

Seven units participated in our study on a voluntary basis. The kick-off meeting of the project gave the opportunity to update clinical knowledge and governance policies. The site visits established personal relationships with the experts and from the discussion of data the specific weak points of each unit could be identified to set individual goals.

Data collected before the intervention from 2901 patients (Table 1) showed large baseline differences among centers. This testifies to the wide diversity in patients’ case mix and clinical behaviour as reported in Ref. [2]. 

The results of the project were positive for the majority of indicators, apparently without causing patients any harm. As in other ASPs [5,8,9,10,11,13,26,27], there were reductions in antibiotic prescriptions (especially quinolones), treatment duration, and MDR emergence.

As quinolones are considered as facilitators of MDR [28,29,30,31,32], the drastic reduction of their prescriptions confirms the willingness to improve therapeutic strategies based on scientific knowledge and compliance to protocols. Shorter durations of empirical treatments suggest more efficient management of microbiological samples, from withdrawal to reporting of sensitivity tests. The marked reduction of ICU-acquired MDR infections, although globally non-significant and with quite large differences between centers, illustrates a general improvement in the management of infected patients, regarding either antibiotic prescriptions or infection control.

No before/after changes could be found in the appropriateness issues (Table 2). Our expectations were probably too ambitious and the methodology and definitions not able to detect differences in the prescribing behaviour. 

Non-significant changes were observed in the use of carbapenems, and a specific study may be necessary to understand this result more in fully. 

The original plan of the study included one more year of observation in 2020 to test the “survival” of improved clinical practice, but unfortunately the COVID pandemic changed the case mix and the ICU work so deeply that comparisons would be meaningless.

Nonetheless, what have we learned from this experience? The enthusiastic acceptance and collaboration of clinicians delegated by each ICU as project contact persons underline the intensivists’ interest in improving clinical practice. 

In view of the voluntary nature of the project, it was hard to engage colleagues not directly involved in the ASP. In a few ICUs, local site visits were limited because of work shifts, holidays, or lack of interest. The results of the project are more effective and enduring when the ASP message and the need for its implementation are shared among the whole ICU staff.

The ICU is a key node in the complex hospital network of players involved in the management of infections. However, an ASP would not be effective if devoted only to ICU physicians and nurses. For this reason, we also invited on-site microbiologists, pharmacists, and infectious disease specialists to participate at the site visits and encouraged the creation of multidisciplinary teams.

Our pilot project was very resource-consuming: we could never offer it to the approximately 200 units associated with GiViTI. To extend the program to other ICUs, we would have to identify which parts of our project were essential and which could be resized, saving workforce and time.

Furthermore, the medical community has to take account of the terrible impact of the SARS-CoV-2 pandemic on the use of antibiotics in the population, inside and outside hospitals and ICUs [33]. ASPs will be urgently and widely necessary, at least to return to the basic concepts of proper antibiotic prescription of the “pre-COVID” era. Similarly, the reduction in MDR infections needs to be rapidly transferred into real COVID-19 life, since in Italy there has been a significant increase of these pathogens. Hopefully our experience will be helpful. 

### Limitations of the Study

The main limitation of the study is the lack of the second year of monitoring the ASP indicators to see if the positive effects were just a “study-related” benefit or it really changed the use of antimicrobial drugs. Most of our participating units are in northern Italy, and most of them became COVID ICUs with a completely different case mix and organisation.

Moreover, the number of participating ICUs was limited. Unfortunately, at that time, few ICUs met the necessary conditions for participation: interest in the study, use of M3 as the EHR, and integration of M3 with the laboratory. For these reasons, we downgraded our study to a pilot study, which, however, gave a considerable amount of important information.

## 5. Conclusions

Our ASP adopted a multidisciplinary approach involving clinicians, microbiologists, pharmacologists, infectious disease specialists, and data scientists. It successfully reduced antibiotic consumption and MDR, without risking patient safety. Simple indicators, which can easily be updated to the newer drugs and different patient populations, were automatically computed from common EHR, helping to monitor ASP data.

The feasibility and the success of this multicenter ASP should now encourage healthcare policy makers to consider that “where there’s a will, there’s a way”. 

## Figures and Tables

**Figure 1 jcm-11-04409-f001:**
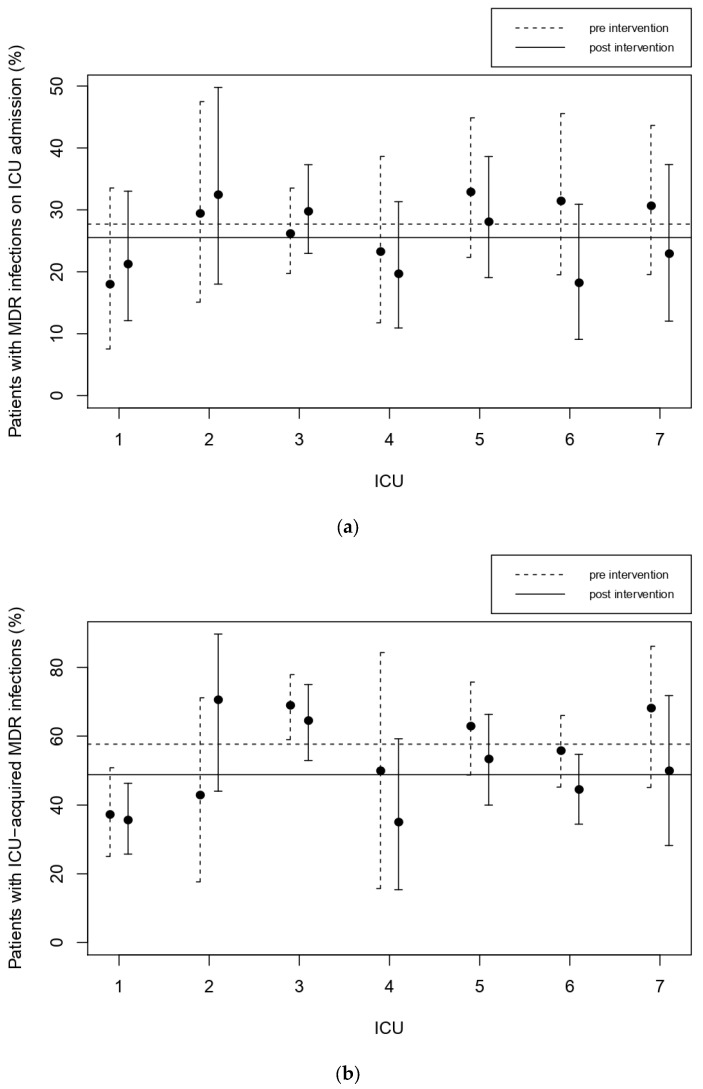
%MDR on admission (panel (**a**)) and %MDR in ICU-acquired infections (>48 h, panel (**b**)) for the participating centers, pre- (dashed line) and post-intervention (solid line). The horizontal line indicates the average.

**Figure 2 jcm-11-04409-f002:**
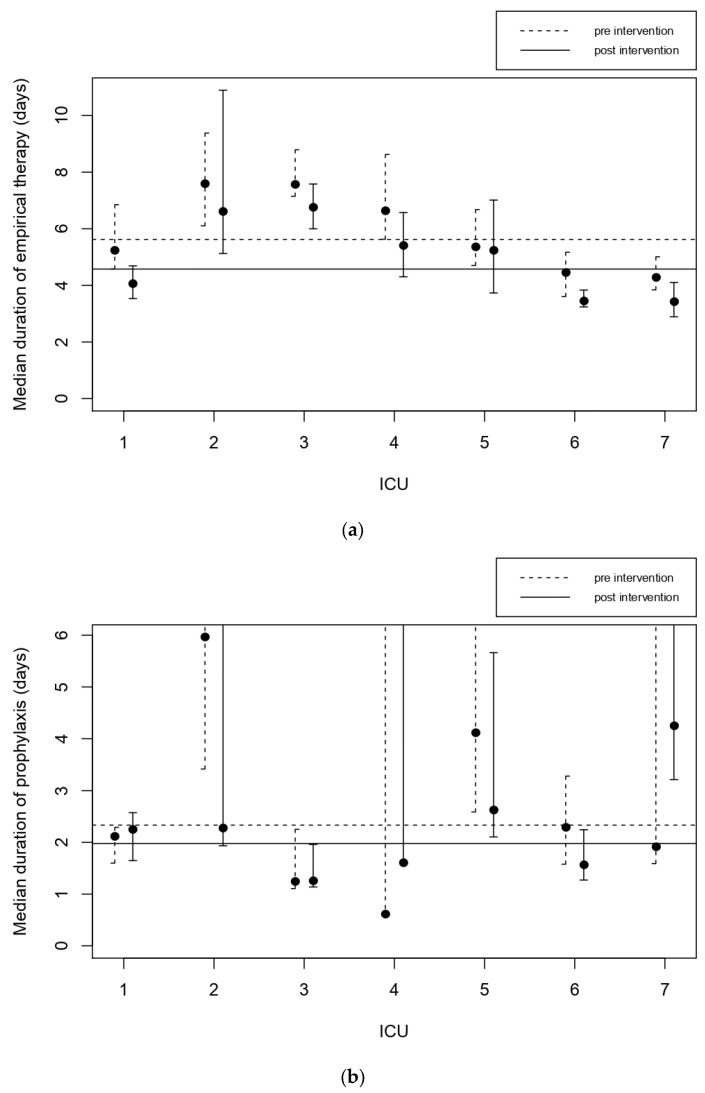
Median duration of empirical therapy (**a**) and prophylaxis (**b**) for the participating centers, pre- (dashed line) and post-intervention (solid line). The horizontal line indicates the average. The use of quinolones more than halved. Before the intervention 15.3% of patients needing antibiotics received quinolones. This decreased to 6.0% after the intervention (*p* < 0.001). Quinolones were used for about 10% to 30% of patients in the seven ICUs. Its usage in all the units decreased in both value and variability, ranging from about 3% to 10% (Figure 3a).

**Figure 3 jcm-11-04409-f003:**
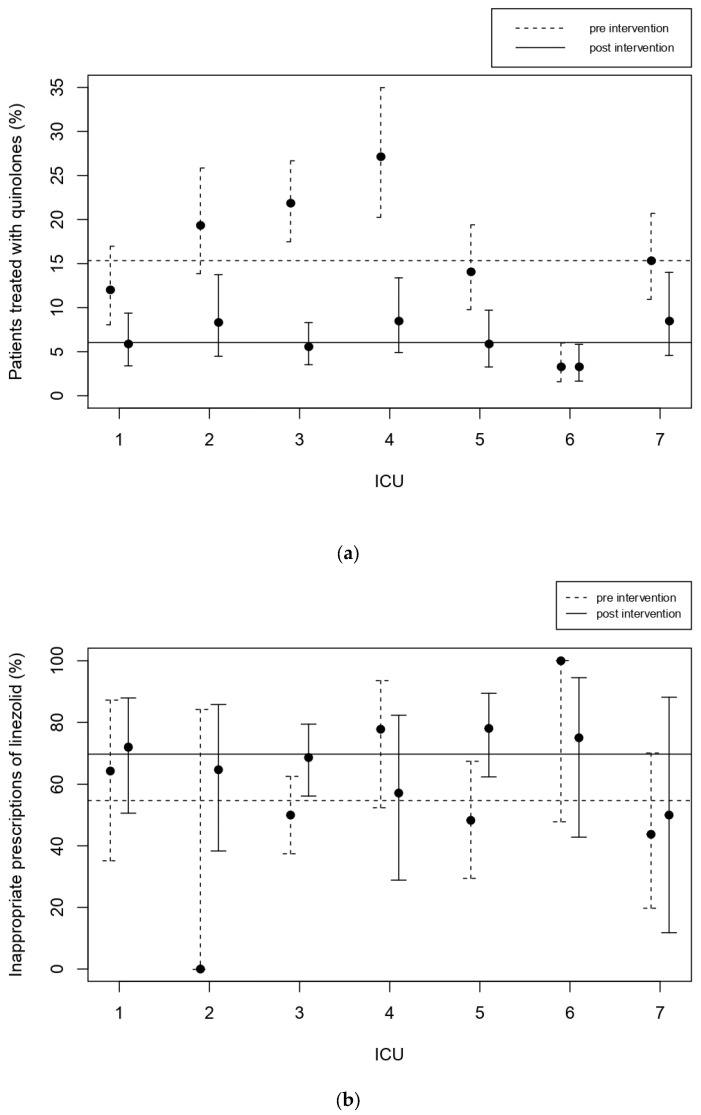
Use of quinolones (**a**) and inappropriate prescriptions of linezolid (**b**) for the participating centers, pre- (dashed line) and post-intervention (solid line). The horizontal line indicates the average.

**Table 1 jcm-11-04409-t001:** Descriptive table (pre-/post-) main demographics, comorbidities, infections present at ICU admission and infections acquired during ICU stay. Significant levels are indicated as * *p* < 0.05, ** *p* < 0.01, *** *p* < 0.001.

	Total (6290)	Pre-Intervention (2901)	Post-Intervention (3389)	*p*-Value	
Median Age (Q1, Q3)	66 (51, 77)	67 (52, 77)	65 (51, 76)	0.003	***
Male	3816 (60.7%)	1755 (60.5%)	2061 (60.8%)	0.80	
ICU Outcome	1011 (16.1%)	471 (16.2%)	540 (15.9%)	0.75	
Comorbidities					
Hypertension	2818 (48.9%)	1321 (48.4%)	1497 (49.4%)	0.48	
Severe Obesity (BMI > 35)	979 (17.0%)	440 (16.1%)	539 (17.8%)	0.10	
Arrythmia	839 (14.6%)	391 (14.3%)	448 (14.8%)	0.64	
Type 2 Diabetes	1018 (17.7%)	460 (16.9%)	558 (18.4%)	0.13	
BPCO	840 (14.6%)	401 (14.7%)	439 (14.5%)	0.81	
Tumor	683 (11.9%)	348 (12.8%)	335 (11.0%)	0.05	*
Myocardial Infarction	531 (9.2%)	241 (8.8%)	290 (9.6%)	0.34	
Moderate/Severe Renal Failure	450 (7.8%)	193 (7.1%)	257 (8.5%)	0.05	*
NYHA 2, 3	450 (7.8%)	208 (7.6%)	242 (8.0%)	0.62	
Vasculopathy	409 (7.1%)	239 (8.8%)	170 (5.6%)	<0.001	***
No comorbidities	1069 (18.6%)	552 (20.2%)	517 (17.1%)	0.002	**
Infections on admission					
Pneumonia	579 (9.7%)	286 (10.6%)	293 (9.0%)	0.04	*
Clinical sepsis	226 (3.8%)	98 (3.6%)	128 (3.9%)	0.56	
Peritonitis	241 (4.1%)	118 (4.4%)	123 (3.8%)	0.24	
Urinary tract infections	116 (1.9%)	50 (1.9%)	66 (2.0%)	0.64	
Skin/soft-tissue Infection	102 (1.7%)	45 (1.7%)	57 (1.8%)	0.81	
No infections	4488 (75.4%)	2009 (74.6%)	2479 (76.1%)	0.16	
ICU acquired infections					
Pneumonia	599 (9.5%)	285 (9.8%)	314 (9.3%)	0.45	
Lower respiratory tract infection	211 (3.4%)	103 (3.6%)	108 (3.2%)	0.43	
Clinical Sepsis	100 (1.6%)	49 (1.7%)	51 (1.5%)	0.560	
Primary bloodstream infection	128 (2.0%)	60 (2.1%)	68 (2.0%)	0.86	
Urinary tract infection	95 (1.5%)	39 (1.3%)	56 (1.7%)	0.32	

**Table 2 jcm-11-04409-t002:** Endpoints with % pre-/post- (aggregated) and *p*-values for all indicators. Significant levels are indicated as * *p* < 0.05, *** *p* < 0.001.

	Pre-Intervention	Post-Intervention	*p*-Value	
Frequency of patients with MDR infections (*N*/*D*)	44.9% (315/701)	39.5% (305/772)	0.11	
On admission (*N*/*D*)	27.7% (131/473)	25.5% (135/529	0.59	
ICU acquired (*N*/*D*)	57.7% (203/352)	48.8% (189/387)	0.09	
Median (IQR) duration of empirical therapy (*D*)	5.6 days (1275)	4.6 days (1406)	<0.001	***
Median duration of prophylaxis (*D*)	2.3 days (589)	2.0 days (584)	0.06	
Inappropriateness of antibiotics by penetration into the site of infection (*N*/*D*)	2.3% (49/2117)	1.9% (49/2619)	0.26	
Inappropriateness of antibiotics by microorganism resistance pattern in empirical therapy (*N*/*D*)	16.2% (57/351)	17.3% (67/387)	0.84	
Inappropriateness of antibiotics by microorganism resistance pattern in targeted therapy (*N*/*D*)	3.8% (19/507)	4.8% (29/606)	0.29	
Use of quinolones (*N*/*D*)	15.3% (251/1637)	6.0% (105/1737)	<0.001	***
Inappropriate prescriptions of carbapenems in empirical therapy (*N*/*D*)	45.2% (19/42)	36.9% (24/65)	0.51	
Inappropriate prescriptions of carbapenems in targeted therapy (*N*/*D*)	36.7% (18/49)	55.3% (42/76)	0.07	
Inappropriate prescriptions of colistin in targeted therapy	27.6% (8/29)	40% (2/5)	0.61	
Inappropriate prescriptions of linezolid (*N*/*D*)	54.9% (82/150)	69.8% (127/182)	0.01	*
Average ICU Length of stay (*D*)	5.5 days (2901)	5.4 days (3389)	0.07	
ICU Mortality (*N*/*D*)	16.2% (471/2901)	15.9% (540/3389)	0.54	

## Data Availability

Data can be accessed upon request and under appropriate data sharing agreement.

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
