# Peer review of "Effectiveness of a Multifaced Antibiotic Stewardship Program: A Pre-Post Study in Seven Italian ICUs"

_jcm, 2022, doi:10.3390/jcm11154409_

Round 1

Reviewer 1 Report

This study enrolled seven italian UCI units in an antibiotic stewardship intervention. There is several types of ASI in the literature and the one presented in this study could be classified as a very elaborated, considering the personnal enrolled. Nevertheless, since the professionals were not present in the UCI on a regular basis, data has to be acquired by each hospital personnal. This could be pose a risk for bias and I think it should be addressed in the discussion.

The authors correctely pointed out that not having a third visit was a limitation. Nevertheless, it would be interesting to have another visit to evaluate if the changes documented persisted and were incorporated in each institution culture. 

Having more points in the time frame would make it possible to use interrupted time series analysis, that would provide a more robust estimate for the changes noted and their persistent over time.

The outcomes were straithfoward and easily to measure. Comparison with other studies will be easier. Nevertheless, the nature of the intervention should be addressed as a limitation, specially because it would be difficult to compare with other studies.

Reviewer 2 Report

Thank you for asking me to review the manuscript entitled ‘Effectiveness of a multifaced antibiotic stewardship program: a pre-post study in 7 Italian ICUs’. Overall this is a well written article and addresses an area of need of establishing antimicrobial stewardship programs in the ICU setting. I also note that this is a pilot study and further studies may be planned in the future.  I have included the following comments/suggestions to improve the readability of the manuscript and application to a wider audience.

Abstract

In line 47 the authors state ‘without risking patient’s safety’. It was not clear from the methods or results how this was evaluated.

Methods

The authors chose a number of indicators to evaluate the success of the antibiotic stewardship program.  

One of the outcomes that was evaluated was the ratio of patients with at least one infection due to MDR bacteria over a year. It is very difficult to show a change in resistance patterns over short time period such as a year. Also it is challenging to demonstrate that the reduction in the use of antimicrobials will result in a concomitant decrease in MDR. This reflects the complexity of resistance emergence, transmission and persistence.  It also highlights the importance of a multifaceted approach to minimising antimicrobial resistance, including infection control management and antibiotic stewardship activities. This should be acknowledged by the authors and discussed in more detail.

The appropriateness of therapy requires further clarification and justification. There have been well described methodologies for assessing appropriateness of therapy and it was not clear why the authors chose the indicators in the manuscript. Bishop JL , Schulz TR , Kong DCM , James R , Buising KL . Similarities and differences in antimicrobial prescribing between major city hospitals and regional and remote hospitals in Australia. Int J Antimicrob Agents 2019;53:171–6 . In addition, the adherence of antibiotic prescribing to guidelines was not evaluated. This should be clarified.

It would be very difficult to assess inappropriateness of antibiotics by penetration into the site of infection. This requires further clarification.

With regards to reviewing appropriateness of therapy it was not clear why the authors focused only on the following antibiotics: fluoroquinolones, linezolid, carbapenems and colistin. This should be clarified.

Also the authors should clarify why they chose to define the appropriateness of therapy as follows: treatment with carbapenems or colistin was considered inappropriate when the microorganism causing the infection was responsive narrower spectrum antibiotics such as penicillins or cephalosporins.  With regards to linezolid it was considered inappropriate in patients with acute renal failure.Therapies in patients with SNC infection by Gram+ bacteria or any infection due to MRSA or VRE were deemed appropriate.

Discussion

Could the authors please clarify the terminology ‘pharmacologists’? Does this refer to pharmacists or medical practitioners who have specialised in the area of pharmacology?

Under limitations it would be worthwhile discussing the ballooning effect of restricting a certain class of antibiotics while prescribers use of another antibiotic class instead of the restricted antibiotic.

Reviewer 3 Report

The study is a comprehensive work emphasizing the importance of an integrated health care system. While reading the result and discussion I wondered if the discussion was correlated with recent similar studies reported in the literature? please see if it is required otherwise ignore. Such studies are imperative to be conducted.
